# Effect of Ti on Characterization and Properties of CoCrFeNiTi_x_ High Entropy Alloy Prepared Via Electro-Deoxidization of the Metal Oxides and Vacuum Hot Pressing Sintering Process

**DOI:** 10.3390/ma16041547

**Published:** 2023-02-13

**Authors:** Hui Li, Sheng Zhang, Jinglong Liang, Meilong Hu, Yu Yang

**Affiliations:** 1College of Metallurgy and Energy, North China University of Science and Technology, Tangshan 063210, China; 2College of Materials Science and Engineering, Chongqing University, Chongqing 400044, China; 3Comprehensive Testing and Analyzing Center, North China University of Science and Technology, Tangshan 063009, China

**Keywords:** high entropy alloys, molten salt electro-deoxidization, mechanical properties, corrosion resistance

## Abstract

The CoCrFeNi system is one of the most important high entropy alloys (HEAs) systems. By adding and adjusting the alloy element components and using different synthesis methods, different phases, organization and microstructure can be obtained, thus improving their properties. In this study, CoCrFeNiTi_x_ HEAs with various Ti contents (x in molar ratio, x = 0, 0.5, 1.0, 1.5) were fabricated by an electrochemical process by virtue of different oxides. The impacts of different Ti contents on the structure, distribution of elements, mechanical properties and corrosion behavior were researched using XRD, EDX and other testing methods. The bulk CoCrFeNiTi_x_ (x = 0, 0.5, 1.0, 1.5) HEAs could be obtained through vacuum hot pressing sintering process (VHPS), which had a single-phase FCC structure. The results of the study showed that the bulk CoCrFeNiTi_x_ exhibited superior ultimate tensile strength (UTS) and hardness, with the UTS of CoCrFeNiTi as high as 783 MPa and the hardness of CoCrFeNiTi_1.5_ reaching 669 HV. The corrosion behavior of CoCrFeNiTi_x_ (x = 0, 0.5, 1.0, 1.5) HEAs in 0.5 M H_2_SO_4_, 1 M KOH and 3.5 wt% NaCl was improved with addition of Ti. CoCrFeNiTi_x_ (x = 0, 0.5, 1.0, 1.5) HEAs have great potential for application in the fields of biomedical coating and aerospace, as well as extreme military industry, etc.

## 1. Introduction

High entropy alloys (HEAs) are a new group of engineering materials with multiple primary elements and interesting mechanical properties what have recently attracted attention in the field of metallic materials research. Their simple crystal structures offer a new way to develop advanced alloys that display tailored mechanical properties, unlike conventional alloys, which are based on one or two major elements [1,2,3,4,5,6,7]. Five or more elements compose high entropy alloys through atomic ratios that are the same or almost the same and which have concentrations of major elements ranging from 5at% to 35at%. The distinctive composition gives HEAs four core effects: sluggish diffusion effect, severe lattice-distortion effect, high-entropy effect and cocktail effect [8]. HEAs tend to form stable solid solution phases and have superior properties, such as high strength, superior oxidation resistance, tensile strength, facture resistance and corrosion. FCC based CoCrFeNi alloys are classified as HEAs because of their high mixed entropy and excellent room- and low-temperature mechanical properties, which have attracted significant academic attention and research [9,10,11,12,13].

Many researches have reported the impacts of various Ti concentration on phase, microstructure and properties of CoCrFeNiTi_x_ HEAs. Research demonstrates that CoCrFeNiTi_x_ HEAs changed from simple FCC structure to σ phase, R phase or Laves phase intermetallic compounds in HEAs with variation of Ti concentration [14,15,16]. CoCrFeNiTi_x_ HEAs fabricated by different preparation methods, such as directed energy deposition high throughput synthesis [17], gas atomization [18], mechanical alloying [19] and electric arc melting [20] showed superior mechanical performance [21,22]. In the meantime, it has been demonstrated that post-treatments, such as aging [23,24], annealing [25] and thermomechanical processing [26,27] can enhance the mechanical performance of HEAs. In addition, there are numerous relevant reports on corrosion resistance [28], magnetic and electrical properties [19,29], and anti-irradiation resistance [30].

Many different methods for the preparation of HEAs have been proposed in recent years, such as: gas atomization [18], mechanical alloying [19] and electric arc melting [20]. The electric arc melting method is used most frequently, but it has the disadvantage of a high experimental temperature, which leads to significant energy consumption. Mechanical alloying is a solid-phase process, but materials produced by this method are susceptible to oxidation, and the cost of raw materials is too high. In the end, the bulk product needs to be prepared by sintering. The gas atomization method is complex and expensive, making it difficult to conduct in a laboratory.

Chen et al. propose the electrochemical approach [31] that has been studied in the fabrication of metal monomers and alloys [32,33,34]. In comparison with classical preparation processes, the fabrication of HEAs by electrochemical process has numerous advantages, such as low temperature, short process flow, and green environment. Electrochemical synthesis of HEAs, such as CoCrFeNi [35,36] and TiNbTaZr [37], via mixed oxides have been demonstrated, but the post-treatment and properties of these products have not been sufficiently studied.

In this paper, CoCrFeNiTi_x_ HEAs powders of various Ti contents (x = 0, 0.5, 1.0, 1.5) were fabricated via electrochemical approach. Bulk CoCrFeNiTi_x_ (x = 0, 0.5, 1.0, 1.5) HEAs were fabricated via VHPS process to investigate the mechanical and electrochemical corrosion properties. CoCrFeNiTi_x_ HEAs are selected for the study because of their excellent mechanical properties.

## 2. Materials Methods

### 2.1. Selection and Proportioning of Raw Materials

In this paper, we are using oxide electro-deoxygenation to prepare CoCrFeNiTi_x_ (x = 0, 0.5, 1.0, 1.5) HEAs, so the mixed powders of CoO, Cr_2_O_3_, NiO, Fe_2_O_3_ and TiO_2_ were selected as raw materials based on their cost and ease of deoxygenation. Table 1 displays the composition of oxide powders for CoCrFeNiTi_x_ HEAs obtained from the atomic ratio calculations.

### 2.2. Electro-Deoxidation Process of Products

The water-free CaCl_2_ was laid in an alumina crucible and desiccated at 573 K for 12 h. The molten salt was pre-electrolyzed at 1173 K with nickel flake as the cathode and graphite as the anode at 2.8 V, and argon gas was continuously injected. Then, the oxide powders were electro-deoxygenated for 8 h at 3.1 V. The products were cleaned by the ultrasonic cleaner and dried under vacuum at 373 K. 

### 2.3. Vacuum Hot Pressing Sintering 

The products were ground and sieved. Then, the CoCrFeNiTi_x_ (x = 0, 0.5, 1.0, 1.5) powders were loaded into graphite molds and hot pressed in a vacuum furnace. The temperature during VHPS was 1373 K, the rate of heating was 30 K/min, and the pressure was 35 MPa for 1 h. Bulk CoCrFeNiTi_x_ (x = 0, 0.5, 1, 1.5) HEAs samples were prepared and tested for their mechanical performance as well as corrosion resistance finally.

### 2.4. Characterization and Test

The electrolytic voltage was provided by a DC power supply (DP310, MESTEK, Shenzhen, China). The phase of bulk CoCrFeNiTi_x_ high-entropy alloy samples with different Ti contents was examined by X-ray diffraction (D/max 2500PC, Rigaku, Japan). The microstructure, the fracture cross-section of bulk HEAs and elemental distribution of samples were characterized by scanning electron microscopy (SEM) and energy dispersive X-ray spectroscopy (EDX) (TESCANVEGA II with Oxford INCA Energy 350). The remaining oxygen contents were measured via an oxygen nitrogen hydrogen analyzer (THC600, Germany). Carbon contents in alloys were analyzed using a carbon sulfur analyzer (G4 Icarus, Germany). The composition of powders was determined by inductively coupled plasma optical emission spectrometer (ICAP6300 DUO, ThermoFisher Scientific, Waltham, MA, USA).

Tensile specimens were polished and ground after hot pressing. Tensile performance was measured using a tensile testing machine with an original strain rate of 10^−4^ s^−1^. Two tests were performed for each sample, and good reproducibility was found. Hardness was measured with a load of 150 g and loading speed of 70 mm/s for 10 s via Vickers hardness tester (HV-115 type, Mitutoyo, Kanagawa, Japan) Corrosion resistance was measured via an electrochemical workstation. (CHI 660, Shanghai Chenhua Instrument Co. Ltd., Shanghai, China). The working electrodes were HEAs samples. The reference electrode was Ag/AgCl and the counter electrode was a platinum sheet. The scanning rate was 2 mV/s.

## 3. Results and Discussion

### 3.1. Electro-Deoxidization of the Mixed Oxides Powders

The chemical composition of the electro-deoxidization products is listed in Table 2 (at% is atomic fractions). Chemical composition is close to the set compositions of CoCrFeNiTi_x_ high entropy alloy. There were two possible reasons for the difference between the electro-deoxidization products composition and the composition of the CoCrFeNiTi_x_ high entropy alloy. First, some oxides had solubility in molten salts which caused the oxides to dissolve in the molten salt. Second, the fine particles produced in the process of electro-deoxidization might pass through the stainless steel mesh and lead to the loss of some oxides particles.

Figure 1 shows the SEM-BSE images and EDX analysis of the CoCrFeNiTi_x_ (x = 0, 0.5, 1, 1.5) HEAs powders. From low-magnification SEM-BSE images (see in Figure 1a–d), the products powders showed clusters composed of nodular particles. The higher magnification SEM-BSE images (see in Figure 1e–h) show that the nodular particle size of the products is bigger because of the increase in Ti contents. EDX composition maps (see in Figure 1i–l) confirmed that the distribution of elements was homogeneous at the micrometer level.

Table 3 shows oxygen content and carbon content of electro-deoxidation products powders. The Ti content of CoCrFeNiTi_x_ (x = 0, 0.5, 1, 1.5) HEAs powders were 0 wt%, 0.5 wt%, 1 wt% and 1.5 wt%. It can be seen from Table 3 that the oxygen content of alloy powders is proportional to the Ti content. The formation of porous metals can be attributed to how the removal of oxygen from solid oxides leaves oxygen vacancies, allowing porous metals to be formed, which has been confirmed in the literature [38,39,40,41,42]. This means that the molar volume of metal (Vm = Mm/ρm) should be smaller than the molar volume of its oxide (Vo = Mo/nρo) where m and o are the metal and oxide, respectively, V is the molar volume, M is the molar mass, ρ is the density, and n is the number of metal atoms in the molecular formula of the oxides. The Vm/Vo ratio of Ti/TiO_2_ was close to 0.63, which was small. However, the Vm/Vo ratio of Ti/TiO was very close to 0.91. TiO was the intermediate phase in the late electro-reduction process of TiO_2_. Therefore, the dynamics of TiO reduction to Ti were difficult because the inherent porosity of the TiO metal layer was small, especially considering the inevitable high-temperature sintering of the metal. This caused an increase in the oxygen content of the electro-deoxidation products powders. As explained in the previous sections, cathodic carbon deposition is caused by side reactions of CO_3_^2-^ and Ca^2+^. There were a number of factors affecting carbon deposition, mainly related to molten salt composition, electro-deoxidation time and reaction temperature. We therefore researched these factors in depth.

### 3.2. Structural and Morphological Characterization of the VHPS Product

The powders of electro-deoxidization products were hot pressed and sintered at 1373 K and 30 MPa for 1 h to obtain bulk CoCrFeNiTi_x_ (x = 0, 0.5, 1, 1.5) alloys. Figure 2 presents the XRD images of the bulk CoCrFeNiTi_x_ alloys with different titanium contents. Three distinct XRD characteristic peaks were found near 2θ of 45°, 50° and 75°, indicating that the main phase of CoCrFeNiTi_x_ (x = 0, 0.5, 1, 1.5) alloys was FCC phase solid solution. The higher mixing entropy of the alloys was the main reason for the formation of FCC solid solution. The presence of Laves phase (Co_2_Ti), R phase (Ni_3_Ti) and σ phase (FeCr) was observed in the ingot of CoCrFeNiTi_x_ alloys prepared by the melting method, which has been confirmed in the literature [43,44,45]. The reason was related to the high temperature of the melting method and the elemental segregation during the solidification of the liquid metal in the melting process. The oxide precursor was deoxidized and alloyed during the electro-deoxidization process. There was no liquefaction of metals, and the process of alloying was dominated by solid phase diffusion, which prevented the formation of other intermetallic compound phases.

Figure 3 shows the SEM-BSE images and EDX analysis of the bulk CoCrFeNiTi_x_ (x = 0, 0.5, 1, 1.5) HEAs. From the low-magnification SEM-BSE images, darkly colored different phases dispersed in HEAs matrix. It can be seen from high-magnification SEM-BSE images that, as the titanium increases, the size of the light phases gradually decreases and the dark phases slowly increases. EDX analysis of the bulk CoCrFeNiTi_x_ HEAs shows that the light phase is dominated by the segregation of Cr and the dark phase is dominated by titanium. The increase of titanium contents led to a gradual decrease in Cr segregation and a gradual increase in titanium segregation. In addition, the segregation size of titanium was small and presented particle dispersion, which was different from Cr segregation aggregation. Carbon deposition at the cathode was the main cause of Cr segregation. Comparison of CoCrFeNi and CoCrFeNiTi_0.5_ revealed the same distribution of C and Cr elemental segregation, which could be explained by the formation of Cr_7_C_3_ between deposited C and Cr during hot-press sintering. Comparison of CoCrFeNiTi and CoCrFeNiTi_1.5_ revealed the same distribution of C and Ti segregation, which meant that TiC was formed between deposited C and Ti during hot-press sintering. With the increase of Ti, the C in the alloys changed from Cr_7_C_3_ to TiC because C tended to form a TiC phase with Ti. In their experiments, Y.B. Peng et al. [46] found that the source of C could also be the graphite mold due to the high sintering temperature as well as the long VHC process; these factors can cause carbon contamination in the graphite mold.

### 3.3. Mechanical Properties and Corrosion Behavior of the VHPS Products

#### 3.3.1. Tensile and Hardness Results

Figure 4 shows the stress–strain curves of the CoCrFeNiTi_x_ (x = 0, 0.5, 1, 1.5) HEAs. It can be observed from Figure 4 that the UTS of the CoCrFeNiTi_x_ HEAs is closely related to Ti. In general, increasing Ti contents was more beneficial for improving the UTS of the HEAs, but their relationship was not a monotonic increase. The UTS of the HEAs changed from 670 MPa (Ti_0_) to 780 Mpa (Ti_1_). However, the UTS of the HEAs changed from 780 MPa (Ti_1_) to 460 MPa (Ti_1.5_). Therefore, the UTS of CoCrFeNiTi_x_ alloys showed a phenomenon of first increasing then later decreasing as the Ti content increased. Solid solution strengthening of the alloy was the main factor for the increase in UTS of the HEAs. The small amount of C deposited in the powder sample generates TiC with Ti during the hot pressing and sintering process. The phase stabilized TiC second-phase plasmas were diffusely distributed in the alloy matrix, which played the role of second-phase strengthening and also contributed to the increase of UTS of CoCrFeNiTi_x_ alloy. The elongation of the CoCrFeNiTi_x_ alloy also showed a phenomenon of first increasing (Ti_0_~Ti_0.5_ alloy) then later decreasing (Ti_0.5_~Ti_1.5_ alloy) as the Ti content increased. Ti_0.5_ alloy had excellent elongation at break compared to other CoCrFeNiTi_x_ (x = 0, 0.5, 1, 1.5) HEAs.

As is commonly known, the Ti-O chemical bond was stable; this made the deep deoxygenation of TiO_2_ exceptionally difficult and resulted in an increase of the residual unremoved oxygen content in the product alloy powder as the Ti content increase. The residual oxygen formed a secondary TCP phase in the matrix. Stress concentration and cracking were induced during the tensile process, and the reason for the significant decrease in alloy toughness was the accumulation of interfacial dislocations caused by the large asymmetry between the TCP phase and the FCC phase [47]. Figure 5 presents the fracture surfaces of CoCrFeNiTi_x_ HEAs. The dimple morphology was exhibited in Ti_0_ and Ti_0.5_ HEAs (see Figure 5a,b) because the FCC phase was the main physical phase of the CoCrFeNiTi_x_ HEAs; this caused it to exhibit significant toughness fracture. The increased oxygen contents of the HEAs caused the Ti_1.0_ alloy and Ti_1.5_ alloy samples to show river pattern detachment fracture morphology that is typical of brittle fracture.

Figure 6 shows the hardness of the tensile test samples. The hardness of CoCrFeNiTi_x_ HEAs gradually increased as Ti increased, from 245 HV in Ti_0_ alloy to 669 HV in Ti_1.5_ alloy. There were two explanations for this phenomenon: first, the Ti atoms with larger atomic radius caused lattice distortion in the high-entropy alloy, and the lattice distortion became more serious because of the increase of Ti contents [48]; second, carbon and Ti generate hard TiC second-phase particles, and the hardness of the alloys was significantly increased by the strengthening effect of second-phase particles.

#### 3.3.2. Potentiodynamic Polarization

Figure 7a displays stable polarization curve of CoCrFeNiTi_x_ alloy at 0.5 M H_2_SO_4_. It can be observed from Figure 7a that a passive region was formed in the total curves range of −0.3–0.9 V_SHE_. All HEAs samples showed a secondary passive region, and the difference in the range of passive region of the HEAs samples was not significant. It can be seen from Table 4 that Ti can significantly improve the E_corr_ of the HEAs, but the increase of E_pit_ is not obvious. The corrosion current density was lower than 304 stainless steel and pure titanium at 0.5 M H_2_SO_4_ [49], indicating that the addition of Ti could promote the formation of corrosion-resistant high-performance oxide films. Ti could easily form an oxide film on metal surfaces in 0.5 M H_2_SO_4_. The oxide film had excellent acid corrosion resistance, and the thickness had little significant effect on corrosion resistance.

Figure 7b shows the stable polarization curve of CoCrFeNiTi_x_ alloy at 1M KOH. Figure 7b shows that, on the surface of the HEA, a clear passive region was formed between the curves and a stable oxide film was formed in 1 M KOH. The passivated region range of Ti_0_ alloy was significantly wider than the passive region of Ti_0.5_, Ti_1.0_ and Ti_1.5_ alloys. As can be seen from Table 5, the Ti_0_ alloy had high E_pit_, and the addition of Ti did not obviously improve the E_corr_ of the alloy, and the i_corr_ was slightly reduced; these items indicate that the corrosion resistance of the HEAs improved slightly with the addition of Ti in 1 M KOH.

Figure 7c shows the stable polarization curve of CoCrFeNiTi_x_ alloy at 3.5 wt% NaCl, and that the passive region range of Ti_1.5_ alloy is significantly narrower than the passive region of Ti_0_, Ti_0.5_ and Ti_1.0_ alloys. This might be because the Ti_1.5_ alloy with higher oxygen content was more prone to localized and pitting corrosion in the oxygen enriched region during polarization tests, where the passivated film was struck through and the extent of the passive region was reduced. The increase in Ti contents leads to an increase in the nano-dispersion of TiN, resulting in the formation of a passive film of TiO_2_ during the corrosion process. The breakage potential of passivation in pitting corrosion increases with the increase of TiN contents in the film [50]. Table 6 shows that the E_corr_ of the HEAs ranged from −1.138 to −1.224 V_SCE_ and the E_pit_ ranged from −0.142 to −0.512 V_SCE_. Ti_0_ and Ti_0.5_ HEAs had higher E_corr_ and E_pit_ because the oxide precursor deoxidization was more difficult with the increase of Ti, and the elevated oxygen content in the product reduced the corrosion resistance of the HEAs. Secondly, the secondary phase in the high Ti alloy also deteriorated the corrosion resistance of the HEAs.

## 4. Conclusions

The CoCrFeNiTi_x_ HEAs with different Ti contents were prepared by electro-deoxidization of metal oxides. The products powders showed clusters composed of nodular particles, and the distribution of elements was homogeneous. The oxygen content of alloys gradually increased as Ti increased. The CoCrFeNiTi_x_ HEAs had an FCC structure after hot pressing. The increase of Ti significantly improved Cr elemental segregation in alloys. With the increase of Ti, UTS of the CoCrFeNiTi_x_ HEAs first increased (Ti_0_–Ti_1.0_) and then decreased (Ti_1.0_–Ti_1.5_). The UTS of CoCrFeNiTi was up to 783 MPa, and the hardness of alloys gradually increased as Ti increased, reaching up to 669 HV. The corrosion resistance results displayed that the addition of Ti significantly improved the acid corrosion resistance of the alloy in 0.5 M H_2_SO_4_ solution; the corrosion resistance of alloys slightly increased with addition of Ti in 1 M KOH solution; Ti_0_ and Ti_0.5_ alloys had higher corrosion potential and pitting potential in 3.5 wt% NaCl solution; and Ti_0.5_ alloy had the best corrosion resistance.

## Figures and Tables

**Figure 1 materials-16-01547-f001:**
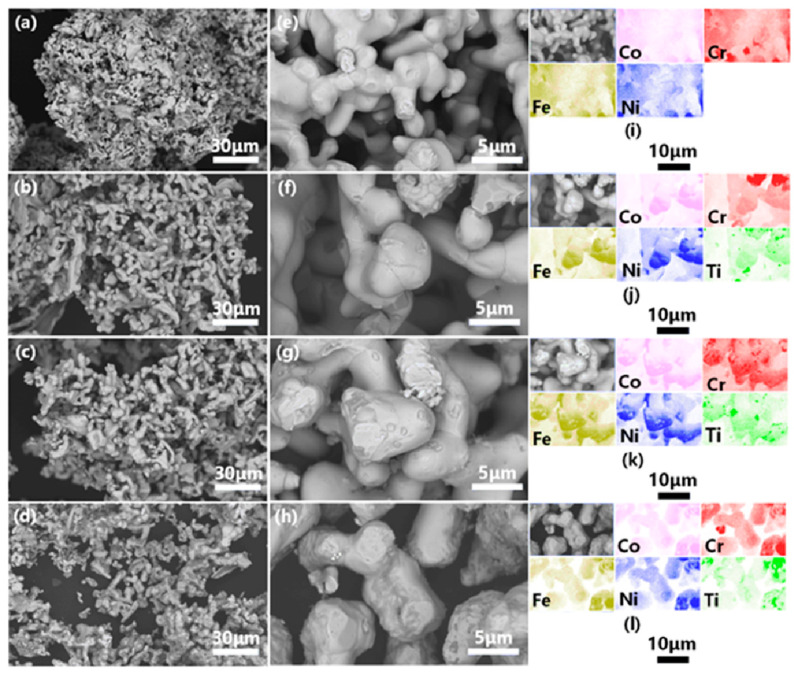
SEM-BSE images and EDX analysis of CoCrFeNiTi_x_ HEAs powders, (**a**–**c**) Ti_0_, (**d**–**f**) Ti_0.5_, (**g**–**i**) Ti_1.0_, (**j**–**l**) Ti_1.5_.

**Figure 2 materials-16-01547-f002:**
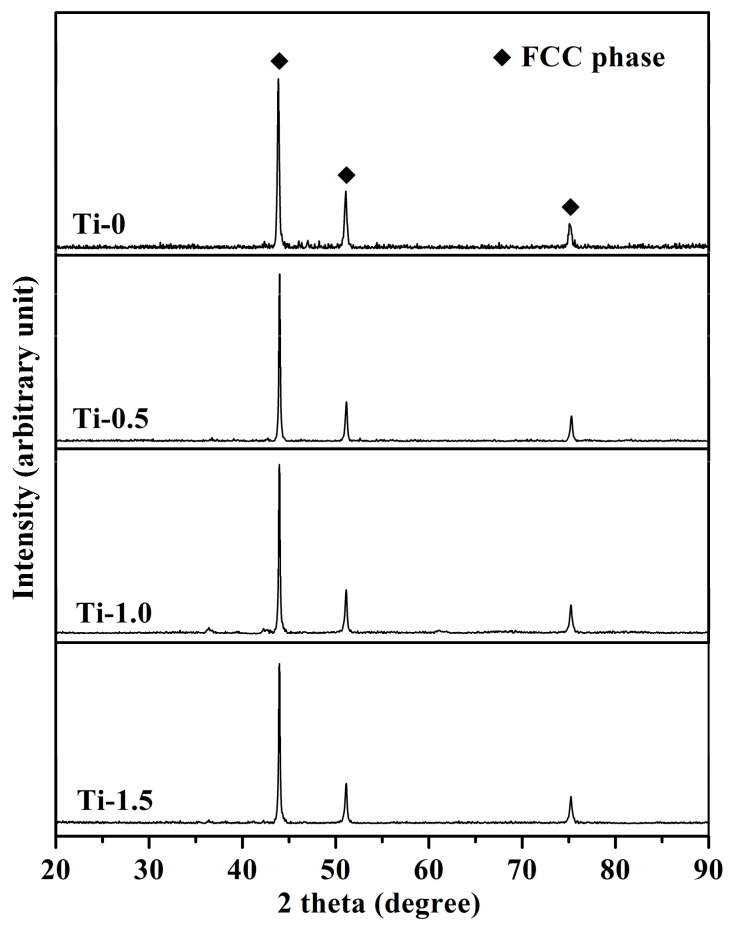
XRD patterns of the bulk CoCrFeNiTi_x_ (x = 0, 0.5, 1, 1.5) HEAs with different Ti contents.

**Figure 3 materials-16-01547-f003:**
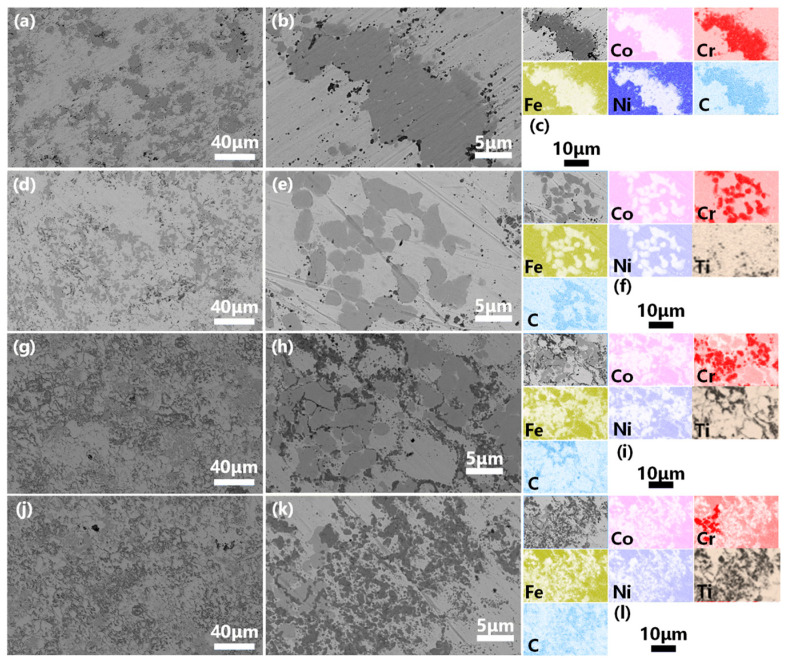
SEM-BSE images and EDX analysis of the bulk CoCrFeNiTi_x_ HEA, (**a**–**c**) Ti_0_, (**d**–**f**) Ti_0.5_, (**g**–**i**) Ti_1.0_, (**j**–**l**) Ti_1.5_.

**Figure 4 materials-16-01547-f004:**
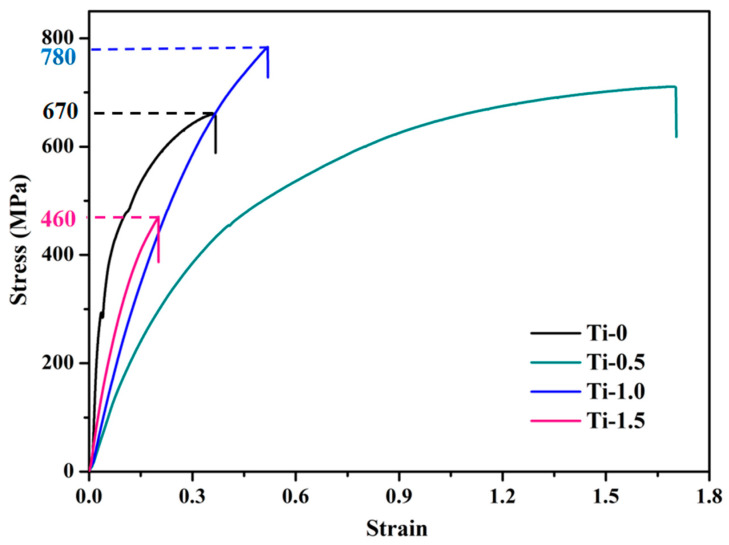
Stress–strain curves of CoCrFeNiTi_x_ HEAs.

**Figure 5 materials-16-01547-f005:**
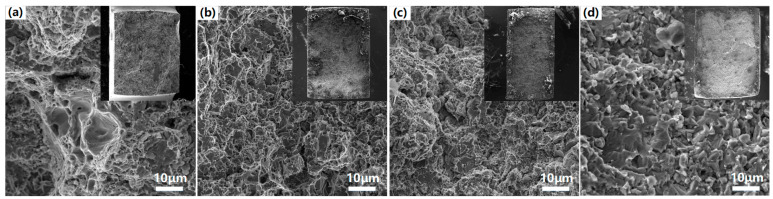
Fracture surfaces of CoCrFeNiTi_x_ HEAs, (**a**) Ti_0_, (**b**) Ti_0.5_, (**c**) Ti_1.0_, (**d**) Ti_1.5._

**Figure 6 materials-16-01547-f006:**
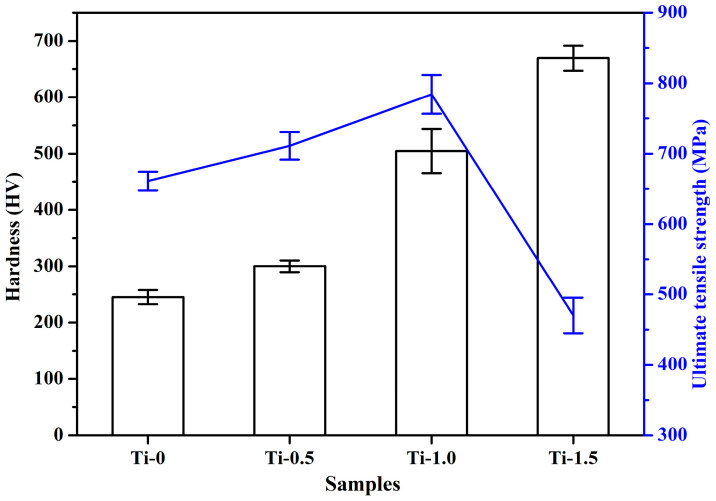
Hardness of CoCrFeNiTi_x_ HEAs and ultimate tensile strength curve.

**Figure 7 materials-16-01547-f007:**
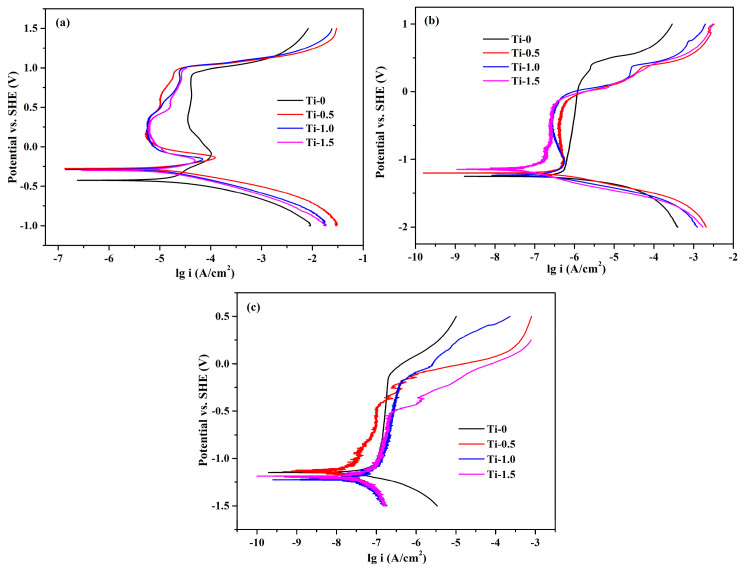
Steady-state polarization curves of CoCrFeNiTi_x_ HEAs (**a**) 0.5 M H_2_SO_4_, (**b**) 1 M KOH, (**c**) 3.5 wt% NaCl.

**Table 1 materials-16-01547-t001:** Composition of oxide powders for CoCrFeNiTi_x_ HEAs (g).

x	CoO	Cr_2_O_3_	Fe_2_O_3_	NiO	TiO_2_	Total Mass
0	3.68	3.73	3.92	3.68	0	15
0.5	3.25	3.29	3.47	3.25	1.73	15
1	2.91	2.95	3.11	2.91	3.11	15
1.5	2.64	2.68	2.82	2.64	4.23	15

**Table 2 materials-16-01547-t002:** Composition of CoCrFeNiTi_x_ HEAs (at%).

x	Co	Cr	Fe	Ni	Ti
0	0.23	0.26	0.27	0.24	0
0.5	0.22	0.23	0.23	0.21	0.09
1.0	0.22	0.19	0.19	0.22	0.18
1.5	0.17	0.18	0.19	0.21	0.25

**Table 3 materials-16-01547-t003:** Oxygen and carbon content of electro-deoxidation products powders (wt%).

Sample	Ti_0_	Ti_0.5_	Ti_1.0_	Ti_1.5_
Oxygen content/wt%	0.35	0.51	0.78	0.93
Carbon content/wt%	1.03	1.15	1.23	1.21

**Table 4 materials-16-01547-t004:** Electrochemical parameters of CoCrFeNiTi_x_ HEAs (0.5 M H_2_SO_4_).

Sample	Solution	E_corr_ (V) vs. SCE	E_pit_ (V) vs. SCE	i_corr_ (A/cm^2^)
Ti_0_	0.5 M H_2_SO_4_	−0.425	0.921	4.26 × 10^−5^
Ti_0.5_		−0.276	0.968	1.39 × 10^−5^
Ti_1.0_		−0.290	0.972	1.48 × 10^−5^
Ti_1.5_		−0.300	0.977	1.64 × 10^−5^

**Table 5 materials-16-01547-t005:** Electrochemical parameters of CoCrFeNiTi_x_ HEAs (1 M KOH).

Sample	Solution	E_corr_ (V) vs. SCE	E_pit_ (V) vs. SCE	i_corr_ (A/cm^2^)
Ti_0_	1 M KOH	−1.251	0.375	2.06 × 10^−7^
Ti_0.5_		−1.202	−0.138	3.02 × 10^−7^
Ti_1.0_		−1.235	−0.083	1.09 × 10^−7^
Ti_1.5_		−1.150	−0.117	5.61 × 10^−8^

**Table 6 materials-16-01547-t006:** Electrochemical parameters of CoCrFeNiTi_x_ HEAs (3.5 wt% NaCl).

Sample	Solution	E_corr_ (V) vs. SCE	E_pit_ (V) vs. SCE	i_corr_ (A/cm^2^)
Ti_0_	3.5 wt% NaCl	−1.147	−0.142	2.45 × 10^−7^
Ti_0.5_		−1.138	−0.153	7.73 × 10^−8^
Ti_1.0_		−1.224	−0.190	1.07 × 10^−7^
Ti_1.5_		−1.185	−0.512	1.39 × 10^−7^

## Data Availability

Data sharing is not applicable to this article.

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
