# Peer review of "Effect of Ti on Characterization and Properties of CoCrFeNiTix High Entropy Alloy Prepared Via Electro-Deoxidization of the Metal Oxides and Vacuum Hot Pressing Sintering Process"

_materials, 2023, doi:10.3390/ma16041547_

Round 1

Reviewer 1 Report

Manuscript ID: materials-2180724

Title: Effect of Ti on characterization and properties of CoCrFeNiTix high entropy alloy prepared via electro-deoxidization of the metal oxides and vacuum hot pressing sintering process

Authors: Hui Li , Sheng Zhang , Jinglong Liang , Meilong Hu , Yu Yang *

Comments: The article is interesting, but there are many gaps that the author must fill. Currently, the article needs major revisions. Authors are required to address these comments for the improvement of the paper.

Why did the authors select these elements only for the mentioned specific application? Many reports are available on CoCrFeNi alloys [Ex: doi.org/10.1016/j.jallcom.2018.09.342; doi.org/10.1016/j.matchar.2020.110807; doi.org/10.1007/978-3-319-48127-2_139 etc.]. What is the novelty of the presented work?

How do authors correlate between mechanical and corrosion? What is the significance of these alloys?

It is better to include a flow chart of the electrochemical process and elaborate on the significance of the adopted method. In section 2, the heading "Experimental Procedure" should be changed to "Materials Methods."

Line number 226, “Ti atoms with larger atomic radius caused lattice distortion…” justify the statement with proper reference.

What is the scientific observation of the alloy samples in Figure 3? It is necessary to explain how Ti is distributed in relation to different concentrations.

Author Response

Hello dear reviewer. Please see the attachment.

Reviewer 2 Report

The manuscript entitled " Effect of Ti on characterization and properties of CoCrFeNiTix high entropy alloy prepared via electro-deoxidization of the metal oxides and vacuum hot pressing sintering process". Authors have reported fabricating CoCrFeNiTix high-entropy alloys with different Ti contents, which shows increasing in corrosion resistance. However, there are some points that need to be corrected. Therefore, recommended the publication of this paper after major revision.

1.      The addresses of the authors and the institution they belong to should be written clearly.

2.      There are deficiencies in the structure of the summary part. Background information about the subject can be given by adding one or two sentences to the first beginning of this section. Experimental work should not be described directly. In the last few sentences of the section, the conclusion of the study should be included. For example, how the study resulted, in which areas it will be used, etc.

3.      Explanations of the abbreviations in the abstract should be included. If the reader reads the first summary, they should be able to understand what these statements mean.

4.      Line 42: Explain the R and σ statements.

5.      There is no information in the introduction about the vacuum hot pressing sintering process (VHPS), which is a part of the study. Is this process a section within other processes, or is it separate? If it is a separate process, please provide information about this process.

6.      The descriptions of the raw materials in line 74 must be added.

7.      Line 129: Specify the image ranges of the EDX.

8.      Line 132: What do BSE images? Write a description of the BSE.

9.      In the data in Table 2, only the effect of the amount of oxygen was investigated. Why is there no comment on the amount of carbon?

10.  Line 193: Edit the punctuation mark in the title.

11.  Line 198: Correct the spelling of Ti0 and Ti1.

12.  In Figure 4, you can add the unit of strain to the graph for integrity or remove the stress in the unit.

13.  Line 238: Correct the spelling of passive.

14.  Line 253-254: The figure number in the text is miswritten. Please make the necessary changes.

Author Response

您好亲爱的审稿人。请参阅附件。

Reviewer 3 Report

Please find the attached file "Comments to the manuscript materials-2180724.docx " where all my comments are presented. 

Author Response

(The authors gave the same response as above.)

Reviewer 4 Report

Reviewer Comment
Manuscript number: Materials- 2180724

Dear Editor,

The manuscript discusses the effect of Ti addition on the mechanical properties of CoCrFeNiTi high entropy alloy. The overall quality of the submitted manuscript will be acceptable after consideration these suggestions:

1-both affiliations are not clear enough belong to whom of the authors.

2-Acronyms should be used consistently throughout the manuscript. For example, HEAs.

3-What does Table S1 mean? Why the authors did not start from Table 1?

4-Table 1, why x repeated in two columns?

5-Table , (at%), is it wt.%, or mol.%, or vol. % or atomic %?

6-Line 150.. Academic expressions should be in articles submitted to high esteem journals such as Journal of Materials.

As we said before .... can be replaced with another term, just for example, As it was explained in section (....) that ......

7-Figure 2, XRD diffractograms,

What are the differences between Ti0, Ti0.5, Ti1.0 and Ti 1.5?

8-in Figure 2, why the authors did not describe the amorphous level of Ti0 compared with that other HEAs? How the Ti addition leads to increase of crystalline of structures?

Author Response

(The authors gave the same response as above.)

Reviewer 5 Report

The manuscript has the following shortcomings. 

1. The manuscript is very poorly written. Some of the sentences are very difficult to understand. For example, Line no. 172: "For micron level microstructure information, example low and high-magnification".

2. Figure 1: Provide high magnification images from Figs. i-l. Is it V or Ti in EDX mapping in Fig. k? The authors should provide the EDX spectra for all the elements measured.

3. Figure 2: Specify the planes, like (111) or (200) in the XRD peaks.

4. Figure 3: Change the color coded for Ti and Cr, as both are RED in color. Again provide the enlarged images and EDX spectra of the Figs. c, f, i, l

5. The reason for the discrepancy between the Hardness and Tensile results.

6. Figure 4. Mention the specific stress and strain in the Figure and the caption.

7. The authors should elaborate the explanations of the mechanical and corrosion properties more extensively comparing with earlier literature. Regarding the passive layer of TiO2 formed during corrosion test, the authors should get more information on the paper, (Structure and properties of Ni1-xTixN thin films processed by reactive magnetron co-sputtering, https://doi.org/10.1016/j.matchar.2020.110604 )
8.  The authors should provide more information on the experimental techniques, characterization and testing methods.  

9. It is highly recommended to provide the optical image of the tensile specimens (before the test) with proper dimension.

Author Response

(The authors gave the same response as above.)

Round 2

Reviewer 1 Report

The revised manuscript can be accepted. 

Reviewer 2 Report

The authors answered all of the comments in the revision file.

Reviewer 3 Report

Dear Authors, 

Thank you for revising the manuscript. All revisions are appropriate, and you responded to all my questions.

I recommend this manuscript to be published in Materials.

Reviewer 5 Report

I feel that the manuscript is up-to-the mark of publication.
